# Terrestrial Laser Scanning: An Operational Tool for Fuel Hazard Mapping?

**Luke Wallace** [1,*,†] , **Samuel Hillman** [2,†] , **Bryan Hally** [3] , **Ritu Taneja** [3] , **Andrew White** [3] **and James McGlade** [3]

1 School of Geography, Planning and Spatial Sciences, University of Tasmania, Hobart, TAS 7005, Australia
2 Department of Environment Land Water and Planning, Melbourne, VIC 3002, Australia; samuel.hillman@delwp.vic.gov.au
3 Geospatial Science, STEM College, RMIT University, Melbourne, VIC 3001, Australia; bryan.hally@rmit.edu.au (B.H.); ritu.taneja@rmit.edu.au (R.T.); andrew.white@rmit.edu.au (A.W.); james.mcglade@rmit.edu.au (J.M.)
* Correspondence: luke.wallace@utas.edu.au
† These authors contributed equally to this work.

**Abstract:** Fuel hazard estimates are vital for the prediction of fire behaviour and planning fuel treatment activities. Previous literature has highlighted the potential of Terrestrial Laser Scanning (TLS) to be used to assess fuel properties. However, operational uptake of these systems has been limited due to a lack of a sampling approach that balances efficiency and data efficacy. This study aims to assess whether an operational approach utilising Terrestrial Laser Scanning (TLS) to capture fuel information over an area commensurate with current fuel hazard assessment protocols implemented in South-Eastern Australia is feasible. TLS data were captured over various plots in South-Eastern Australia, utilising both low- and high-cost TLS sensors. Results indicate that both scanners provided similar overall representation of the ground, vertical distribution of vegetation and fuel hazard estimates. The analysis of fuel information contained within individual scans clipped to 4 m showed similar results to that of the fully co-registered plot (cover estimates of near-surface vegetation were within 10%, elevated vegetation within 15%, and height estimates of near-surface and elevated strata within 0.05 cm). This study recommends that, to capture a plot in an operational environment (balancing efficiency and data completeness), a sufficient number of non-overlapping individual scans can provide reliable estimates of fuel information at the near-surface and elevated strata, without the need for co-registration in the case study environments. The use of TLS within the rigid structure provided by current fuel observation protocols provides incremental benefit to the measurement of fuel hazard. Future research should leverage the full capability of TLS data and combine it with moisture estimates to gain a full realisation of the fuel hazard.

**Keywords:** TLS; fuel structure; risk assessment; fuel hazard; occlusion; remote sensing

## 1. Introduction

In many locations across the globe, climate change is driving a rapid and widespread increase in fire weather conditions conducive to large fires [1–4]. To understand both current and future fire behaviour, land and fire managers seek to model the behaviour of fires and understand the risk associated with a landscape. Fire behaviour is strongly influenced by the properties of vegetation; therefore, an understanding of fuel is a key requirement for the operation of fire simulators.

The physical characteristics and spatial configuration of fine fuels (<6 mm diameter), which are considered to be fuels that readily ignite in a flaming fire front, are of primary consideration for fuel assessments around the world [5]. In South-Eastern Australia, fuel is separated into distinct strata (surface, near-surface and elevated) based on height, with a separate class for bark hazard [6]. These classes are used for modelling where hazard

for a particular area is determined by the vegetation type and time since fire. Visual fuel assessments have also been undertaken to facilitate field assessments where assessors compare the vegetation in a strata to a set of written guidelines and reference images to select an appropriate risk rating based on several metrics, for example coverage of live and dead fuel, vertical continuity, height and density [6,7]. Similar fuel attributes are measured across the world that help to inform fuel dynamics understanding. For example, in the USA, the Fuel Characteristic Classification System is a comprehensive guide where fuel beds are described by firstly identifying the strata and subsequently assessing a number of variables (e.g., cover, height, density) [8]. Cover and height metrics are used both directly and indirectly through the determination of fuel load in fire behaviour models and simulators around the world (U.S.: [9], Canada: [10], Australia: [7,11], Australia and Europe: [12]).

Whilst initially designed to provide a rapid evaluation of the chance that a wildfire cannot be controlled, recently, in-field assessments of fuel hazard have been used to monitor fuel levels in Victoria, Australia [13]. Whilst being the best available method at the time of introduction that balances efficiency and ease of assessment by personnel with limited experience, this method lacks the necessary precision, accuracy and repeatability to measure fuel dynamics and validate fuel models [14–16].

## 1.1. Mapping and Monitoring Vegetation with Terrestrial Laser Scanning

Terrestrial Laser Scanning (TLS), an active Remote Sensing (RS) technology, has been extensively used to capture representations of forest environments and forms the benchmark that emerging terrestrial 3D RS approaches are compared to [17–19]. TLS can be used to provide descriptions of the surrounding environment that are both precise and accurate to assist in understanding the traits of vegetated ecosystems [20–22]. Whilst much of the application of TLS within forests has focused on providing structural measurements of trees that were commonly observed manually in operational contexts [19,23], the application of the technology for characterising the non-tree components of forest structure has also recently been demonstrated [24], for example in observing understorey biomass [25], monitoring growth and change [26,27] and quantifying fire behaviour relevant variable [28–31].

A key development that has led to the consideration of TLS in an operational context is the reduction in price of this technology. New lower-cost laser scanners such as the Leica BLK360 and Faro Focus M70 are making the deployment of multiple scanners within a jurisdiction possible. These scanners offer similar data products at lower cost and with easier operation (reduced scanner weight and simplified user interfaces) [32]. Initial studies have demonstrated adequate capabilities in measuring large features within forests [32,33]. However, these scanners are unproven in the measurement of fine vegetation structure, yet are beginning to be adopted in an operational context within South-Eastern Australia [34]. Further barriers for operational implementation exist through the large amount of time taken to comprehensively capture a plot (approximately 2–3 h per 2500 m$^2$) and subsequently align scans and analyse the data (approximately 4–6 h) [20]. Despite the unparalleled opportunities to measure fuel changes over time and validate fuel inputs through the use of TLS point clouds, operational implementation of the technology will continue to be limited until a viable approach that minimises the amount of capture and processing time is developed.

## 1.2. Aims and Objectives

The aim of this paper is therefore to outline a robust and accessible method for sampling fuel metrics and therein highlight the potential of TLS to be used as an operational fuel hazard observation technology. A case study was designed to provide guidance on TLS procedures that could be implemented operationally within South-East Australian forests. An analysis of the uncertainty within estimates of fuel hazard metrics derived from point clouds captured from TLS was completed for different sample designs. As such, the research was designed to achieve the following objectives within the context of South-East Australian eucalyptus forests:

1.  To provide guidance as to an appropriate sampling strategy for use in the collection of TLS data;
2.  To compare two Terrestrial Laser Scanners with different operating principles and price points for fuel hazard assessments.

The achievement of these aims would advance the applicability of TLS to be used by land managers to precisely assess fuel metrics at the time of assessment and monitor fuel dynamics over time.

## 2. Materials and Methods

### 2.1. Study Areas

For this study, two sites were selected in Victoria, Australia, and plots of 20 × 20 m in size were established for data collection (Figure 1a). These two sites are located in forest types that are commonly found across South-Eastern Australia. As such, they provide a suitable environment upon which to develop guidance on an appropriate sampling strategy for use in the collection of TLS data within this region. The first site (S1) was at Murrundindi (75 km North East of Melbourne), located in Taungurung Country, in vegetation characteristic of Ecological Vegetation Class (EVC) 18: Riparian forest (Figure 1b). The overstorey at this site was dominated by *Eucalyptus obliqua* (Messmate Stringybark) and *Eucalyptus viminalis* (Manna Gum) that were approximately 30–40 m tall. Understorey trees present at this site were approximately 5–10 m tall and consisted of wattles with *Cassinia aculeata* (Common Cassinia) also occurring across the site. The understorey featured a variety of grasses, sedges, herbs and ferns with *Pteridium esculentum* (Austral Bracken) also present.

The second site (S2) was located at Clonbinane (approximately 50 km North of Melbourne), located on Taungurung Country, in vegetation characteristic of EVC 23: Herb-rich Foothill forest (Figure 1c). Plots within this area were impacted by the 2009 Black Saturday bushfires [35]. The overstorey consisted of medium to tall (20–30 m) *Eucalyptus obliqua* (Messmate) and *Eucalyptus viminalis* (Manna gum). Understorey trees (approximately 5 m) including *Acacia melanoxylon* (Blackwood) and *Acacia dealbata* (Silver wattle) were also present. Understorey shrubs consisted of a mixture of forbs and grasses (e.g., *Pteridium esculentum* (Austral Bracken)) with vegetation in the layer approximately 1–2 m in height.

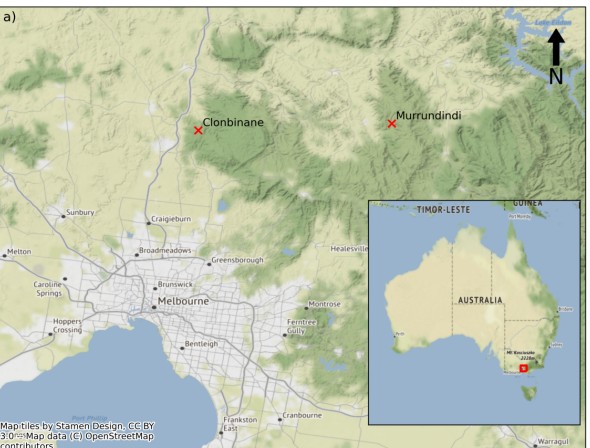
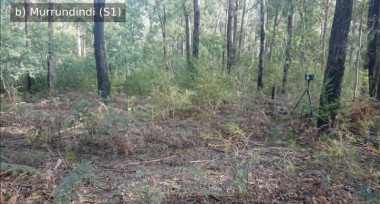
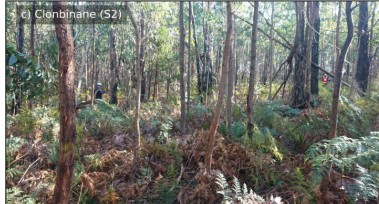

**Figure 1.** (**a**) Locations of the two sites (shown as red crosses) relative to Melbourne, Australia and example photographs of typical vegetation found at (**b**) Murrundindi (S1) and (**c**) Clonbinane (S2) plots.

### 2.2. Fuel Hazard Framework

At each plot, an overall fuel hazard assessment following the method outlined in Hines et al. [6] was undertaken. This included observing the surface, near-surface, elevated and bark fuel classes and their respective characteristics. The results of this assessment are

provided in Appendix A. Pictures were also captured at the time to ensure post-validation could occur (see Figure 1).

### 2.3. TLS Instruments

Two scanners, a Trimble TX-8 and a Faro M70, were used within this study and represent two different ranging technologies (Table 1). The Trimble TX8 laser scanner, with an approximate cost of AUD 110,000, is a scanner operating on the time of flight ranging principle. The time of flight method of capture utilises the time for a laser pulse emitted from the sensor to reflect off a target and return to the sensor, to provide a precise range to that target [36]. Combining this range information with the direction the laser was emitted allows for the 3D location of the target to be determined relative to the sensor [36]. The Trimble TX8 scanner settings were set to capture Level 2 quality scans (11.3 mm point resolution at a distance of 30 m) for an average scan time of 3 min and 20 s.

The FARO M70 laser scanner represents an affordable scanner at approximately AUD 20–30,000. The scanner utilises a phase-shift method of range measurement. Similar to time-of-flight systems, this system also uses an emitted laser pulse, however with phase-based systems the range is determined by the phase difference between the sent and received waveforms. Scan data were captured using a FARO M70 using settings of 1/2 of full capture scanned 2x with 3.1 mm point resolution at a distance of 10 m, for an average scan time of 7 min and 40 s.

**Table 1.** Specification of the TLS (FARO M70 and Trimble TX8) used to capture point cloud information in this study.

| Property | Faro M70 | Trimble TX8 |
|:---:|:---:|:---:|
| Operating Principle | phase-difference | time of flight |
| Wavelength | 1550 μm | 1500 μm |
| Max Range | 70 m | 120 m |
| Ranging Error | 3 mm @ 10 to 25 m | <2 mm @ 2 to 120 m |
| Angular Accuracy | not specified | 0.080 mrad |
| Divergence | 0.3 mrad | 0.177 mrad |

### 2.4. TLS Data Capture

TLS capture of the plot involved sampling in a 5 × 5 m grid encompassing the 20 × 20 m plot area, for a total of 25 scans per plot. The position of the measurement centre of the two scanners at each location was set to be as close as possible to avoid introducing differences in occlusion patterns. Scans were taken from a height of approx 1.7–1.8 m from the ground surface to maximise near field (<5 m from the scanner) capture of near-surface fuel features.

All scans were co-registered in Trimble RealWorks 11.1 (Trimble Inc., Sunnyvale, CA, USA) using a cloud to cloud approach with an accuracy of 10 mm. Five spherical targets were placed within each plot and used to confirm that the accuracy of the final co-registration was sufficient. Individual co-registered point clouds were then exported for further processing.

The merged TX-8 scan at each site was then manually checked and edited for noise. This was completed by extracting ten 1 m wide transects across the plot and removing any points deemed to be not attached to recorded vegetation or the ground. This point cloud was processed in the same manner as the individual scans and used as a point reference for inter-comparing the outcomes from the two scanners.

### 2.5. Point Cloud to Fuel Hazard

Point cloud processing and analysis was completed using Python code that was developed in-house specifically for TLS point cloud processing. The following steps were undertaken.

### 2.5.1. Voxelisation

In order to provide a discrete representation of the environment, point clouds were translated into 3D voxels. A voxel model for these purposes is a three-dimensional grid domain, with the size of the grid cells determining the resolution of the 3D grid [37]. In this case, all point clouds were transformed into 2 cm resolution voxel spaces (with each voxel having a volume of $8 \, cm^3$). This was completed by attributing each point to a voxel, and considering any voxels containing at least one point to be filled.

### 2.5.2. Noise Detection

The rate of noise in a single scan was determined through comparison to the reference scan. As a complete representation of the fine fuel structure at the scale of the plots here is beyond logistical possibilities, the manually edited reference scan is therefore assumed the best representation of the fine fuel structure in the plot. This assumption is made based on previous studies which have shown correlation between TLS measure of fine fuel and point based intercept methods collected at the scale of quadrats [28,30,38,39]. Any voxel that was considered filled in the single scan and not within 6 cm of a filled voxel within the reference scan was considered to be noise due to misregistration errors or environmental effects (such as the vegetation movement due to wind). Whilst these points could be spurious noise points, there is a strong likelihood that they could have originated from vegetation and as such are not considered noise here.

Noise was detected using the filter outlined in Rusu et al. [40]. In brief, the filter assumes noise points are those that are isolated from other points within the point cloud. These points are found by determining the nearest neighbour distance between each point and other points in the point cloud. The mean and standard deviation of the nearest neighbour distance are then calculated. Points are then considered noise if their nearest neighbour distance is greater than the mean distance plus the standard deviation multiplied by a threshold (ST), with this threshold and the number of neighbours (NN) used in determining the nearest neighbour distance set by considering the properties of the scene and sensor.

The filter parameters, NN and ST, were set depending on the 3D distances of the voxel to the scanner. This noise filter was run with NN set from 2 to 40 in steps of 2 and ST set at 2 to 10 in steps of 0.2. Correct parameterisation was determined by maximising the Mathews Correlation Coefficient (MCC) of the filter for 2 m width bins of three randomly selected scans from each site. From this process a single set of parameters were determined for each scanner and used to remove the noise in each individual scan.

### 2.5.3. Ground Definition and Normalisation

The Cloth Simulation Filter (CSF) outlined in Serifoglu Yilmaz et al. [41] was used to determine which voxels originated from the ground. In order to determine the parameters, the CSF filter was run using a range of parameters. The resultant ground voxels were then manually inspected and the best representation of the ground subjectively determined. This was then used to create a reference ground DTM based on Triangular Irregular Network (TIN) interpolation.

Three randomly selected single scans from each site were then filtered using CSF, and a parameter set run for each variable. The parameterisation of the ground was then determined as that with the lowest Root-Mean-Squared Error(RMSE) when compared to the reference DTM. The final parameters were chosen to be grid resolution of 0.24 m, class threshold of 0.03 m, time step of 0.6 s, iterations set to 1000 and rigidity of 3. The resultant DTM was then used to determine the Above Ground Height (AGH) of every voxel, where voxel height was determined as the difference between the absolute height of the voxel being normalised and the absolute height of the DTM at that horizontal location.

### 2.5.4. Establishment of Strata

In order to replicate the strata based observations used in the OFHAG, the layer pouring algorithm presented in Hillman et al. [29] was used. This approach aims to define objects within voxel space based on vertical connectivity. Vertical voxel layers are traversed from the top down. In each layer all horizontally connected voxels are considered a segment. If this segment overlaps (horizontally) with a segment from the above layer they are considered as being from the same object. To avoid a reliance on the complete 3D representation of the environment, the voxel space is diluted using a sphere shaped structuring element with a diameter of 3 voxels.

Beside the voxel dimensions, determined based on the spacing of TLS points at the maximum observation distance, the only parameter required for this algorithm is the size of the structuring element applied. This parameter depends both on the spacing of TLS points and the properties of the environment. In particular, the amount of occluded space within a vertical column. To determine this parameter spherical elements with diameters of 3, 5 and 7 voxels were trialled. The results of the segmentation were visually assessed with a diameter of 3 determined for merged voxel space and 5 for the voxel space of individual scans.

The fuel hazard assessment approach used within the jurisdiction of the study divides a fuel into strata. Whilst not explicitly defined within the guide, height thresholds consistent with fuel studies in eucalypt forests were applied to separate strata. This created four classes which capture the surface and near-surface combined fuel layer ($<0.6$ m), elevated ($0.6$ m to $3$ m), intermediate ($3$ m–$5$ m) and canopy ($>5$ m) fuel layers [6,7]. As the separation of the objects from the surface strata and the ground remain difficult to achieve within TLS [29], we provide an analysis for the derivation of a fuel layer which combines some elements of the surface (those adequately separated from the ground) with the fuel elements from the near surface layer. Estimates of bark fuel were not considered, as currently no method exists to achieve the classification of bark fuel type from TLS data. Objects derived from the layer pouring process were assigned to the layer based on the highest included voxel.

### 2.5.5. Metric Derivation

The primary metrics derived from the TLS point clouds are mean height and cover. Mean height was calculated as the mean height of the top surface of each fuel layer. The top surface was determined as the highest point within each voxel column. Cover was determined as the number of voxel columns containing at least one voxel from each layer divided by the number of voxel columns containing any filled voxels (ground, fuel or noise). In the case of the combined surface and near-surface layers the number of columns containing a voxel from this strata was divided by the number of columns containing any voxels from the ground, surface and near-surface and elevated layers.

### 2.6. Assessing TLS Performance

Based on discussion with local land managers, it was deemed that the most likely method of deployment of TLS scanning in current operational settings is the use of individual scans, or a small number of merged scans. The justification for this was the lower in-field cost in determining suitable scan locations such that overlapping space was adequately captured, as well as lower in-office costs without the need for scan co-registration in post-processing.

As scan alignment requires careful placement of the scanner and extra processing time, the performance of the TLS was based on the ability of individual scans to characterise the surrounding area. In this scenario the number of scans collected to characterise a plot or a landscape is likely dependent on how much information is collected in the scan and the sampling effort required. As such, the focus of this assessment was on how well single scan data characterises the environment at different horizontal distances from the scanner.

The first point of comparison was through the effective measurement rate at increasing horizontal distances from the scanner. The effective measurement distance was determined

through the rates of omission and commission at 0.5 m distance bins from the scanner, where the rate of commission is analogous to the rate of noise and was determined both for noisy and noise filtered point clouds. The rate of omission summarises occlusion as well as other sources of no returns (such as incorrect noise filtering and non-return interceptions). The effective measurement distance was summarised for all voxels, fuel voxels and ground voxels. RMSE was calculated with increasing distance utilising both the DTM determined from known ground points and the DTM determined from CSF ground points.

Metric Evaluation

Rowell et al. [28], Hillman et al. [38], Hudak et al. [42], Loudermilk et al. [43], Cooper et al. [44] demonstrated a high level of accuracy between fine-scale volume and structural measurements derived from TLS and laboratory and field measurements of biomass and fine fuel properties. Therefore, in the context of this case study, the merged TLS scan derived from the Trimble TX-8 is considered to provide the source of validation. Whilst Wallace et al. [27] showed that repeat measurements of an environment can provide high precision estimates of change within an environment following fire.

RMSE (in comparison to the reference point cloud) was used to summarise the differences in fuel hazard metrics described in Section 2.2. For height and layer distance RMSE was calculated for each voxel column, while RMSE for cover was calculated based on the cover for each 50 cm ring extending out from the location of the scanner.

Individual scans were compared to fuel hazard metrics from the reference voxel space at four horizontal radii from the scanner (2, 4, 6 and 8 m). Differences were summarised for both cover and height metrics using the RMSE between these two sets of metrics.

## 3. Results

### *3.1. Merged Scan Plot Characterisation*

#### 3.1.1. Vertical Point Distribution

The point clouds captured by both scanners provided similar overall representation of the two plots Figure 2. The FARO M70 scanner returned a greater number of points ($1.4 \times 10^9$ at S2 and $1.5 \times 10^9$ at S1) in comparison to the TX-8 scanner ($1.1 \times 10^9$ at both sites). Once transformed into voxel space this translated to $8 \times 10^6$ more filled voxels within the Faro M70 representation.

From the vertical distribution of S2 shown in Figure 2b,d, an increase in filled voxels at around 5 m AGH suggest a significant sub canopy layer, whilst the maximum height observed (24.90 m (Faro M70) and 24.89 m (Trimble TX8)) and low density of filled voxels above the sub-canopy highlight a short and sparse canopy which was consistent with in-field observations of dead standing trees above the sub canopy layer. At S1, (shown in Figure 2c), the canopy layer, between 20 and 35 m provides the greatest proportion of vegetated volume.

Only minor differences were seen in the raw distribution of point heights and the distribution of AGH from the voxel space throughout the profiles (Figure 2). The primary difference between the two sensors was seen in the vertical space between the sensor height (approximately 1.7 m AGH) and the ground where the Faro M70 returned a greater proportion of its overall point count. This translated to an observable difference in the proportion of filled voxels in this area (between 0 and 2 m AGH).

#### 3.1.2. Metric Evaluation

The fuel hazard characterisations of the two plots using both scanners were found to be very similar (Table 2). The most significant difference between the two scanners was in the near surface cover estimate at S1. The M70 scanner estimated this cover to be 10% higher than the TX-8. Only a 1% difference was seen in the cover of this layer a S2. The elevated layer was estimated to have a 3% higher cover by the Faro M70 scanner at this site.

The mean height of each layer was found to be estimated similarly by both scanners. With difference of 1 cm in the near surface layer, 1 to 5 cm in the elevated layer and 5 to 9 cm

in the canopy layer. The canopy height at S1 showed the greatest difference (0.38 m of any layer at both sites. The difference in canopy height was only 7% at S2 site. S2, however, had a much lower canopy, 11.9 m in comparison to 29.4 m (as observed by the TX-8 scanner).

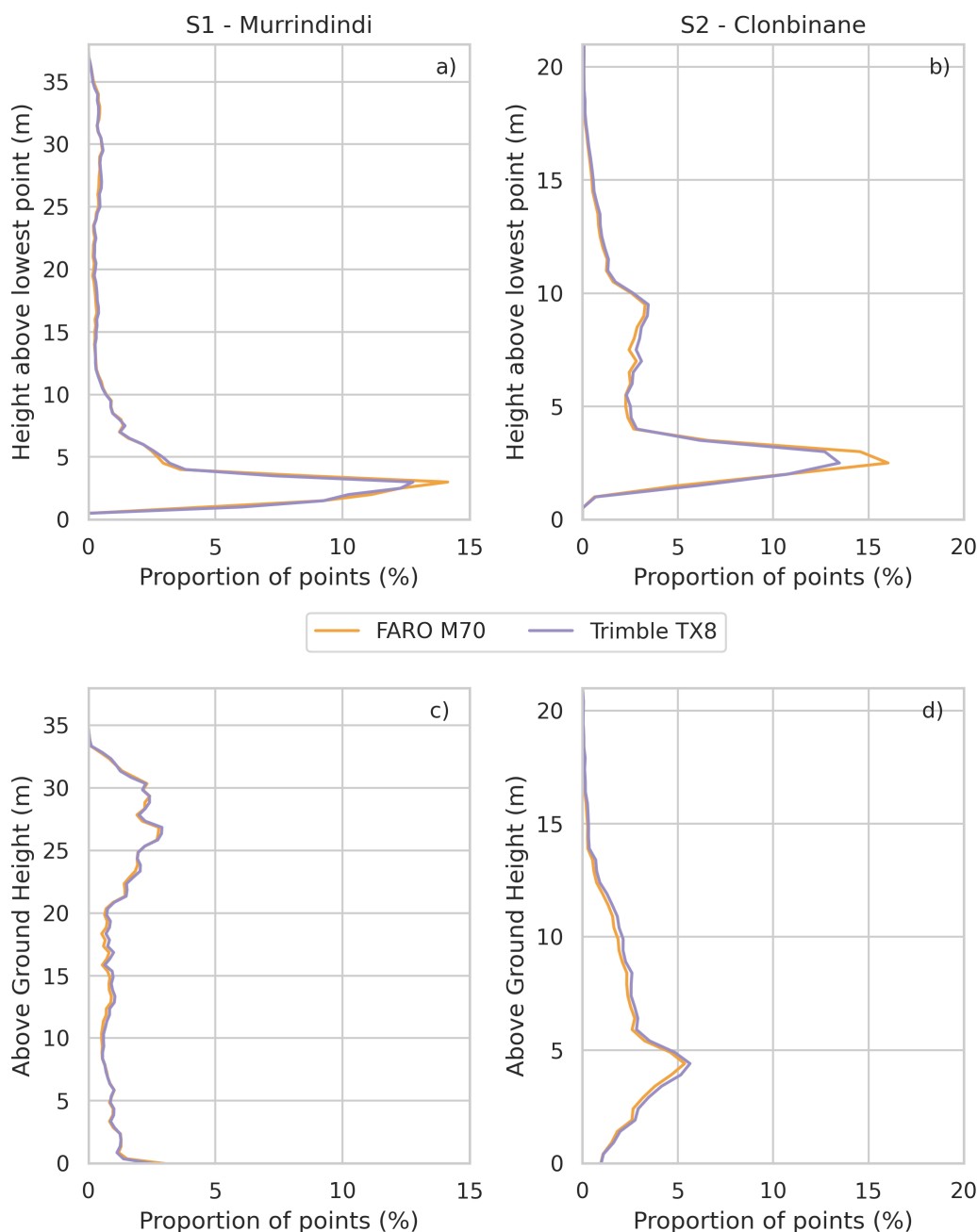

**Figure 2.** Height distribution of point clouds captured by the M70 (green) and TX-8 scanners (pink) at (**a**) S1—Murrundindi and (**b**) S2—Clonbinane relative to the lowest returned point. Above ground height distributions of filled voxels derived from these point clouds at (**c**) S1 and (**d**) S2.

The height estimates derived from the scanners were lower than the visual based assessments in all strata (Table A1). In contrast, the cover estimates derived from the scanners were generally higher than the visual based assessments in all strata. The largest discrepancy was seen in the near-surface layer where estimates were between 10–20% different between the scanner and visual based assessments.

The distribution of volume (or filled voxel count) was also similarly represented by each scanner at both sites (Figure 3). The Faro scanner estimated a greater proportion of

the biomass to be present in the near surface and elevated than the TX-8 scanner at both sites. This added volume contributed to the greater cover estimated in these layers.

**Table 2.** Height ($\mu$: mean, and $\sigma$: standard deviation) and cover of each fuel layer produced by the merged scans from each scanner at the two sites.

| Site | Layer | Scanner | | | | | | Visual Assessment | |
|---|---|---|---|---|---|---|---|---|---|
| | | Trimble TX-8 | | | Faro M70 | | | | |
| | | Height (m) | | Cover | Height (m) | | Cover | Height (m) | Cover |
| | | $\mu$ | $\sigma$ | % | $\mu$ | $\sigma$ | % | | % |
| S1 | NS | 0.15 | 0.15 | 50 | 0.14 | 0.14 | 60 | 0.20–0.40 | 20–40 |
| | E | 1.23 | 0.80 | 36 | 1.18 | 0.81 | 36 | 0.60–0.70 | 20–30 |
| | SC | 4.99 | 1.18 | 24 | 4.94 | 1.17 | 24 | N/A | N/A |
| | C | 29.38 | 4.62 | 88 | 29 | 4.8 | 86 | 25 | 71–80 |
| S2 | NS | 0.20 | 0.15 | 54 | 0.21 | 0.15 | 55 | 0.20–0.40 | 20–40 |
| | E | 0.90 | 0.61 | 48 | 0.91 | 0.59 | 51 | 0.50–0.60 | 30–50 |
| | SC | 5.53 | 1.10 | 40 | 5.44 | 1.13 | 37 | N/A | N/A |
| | C | 11.95 | 4.01 | 63 | 11.89 | 3.97 | 61 | 10 | 60–70 |

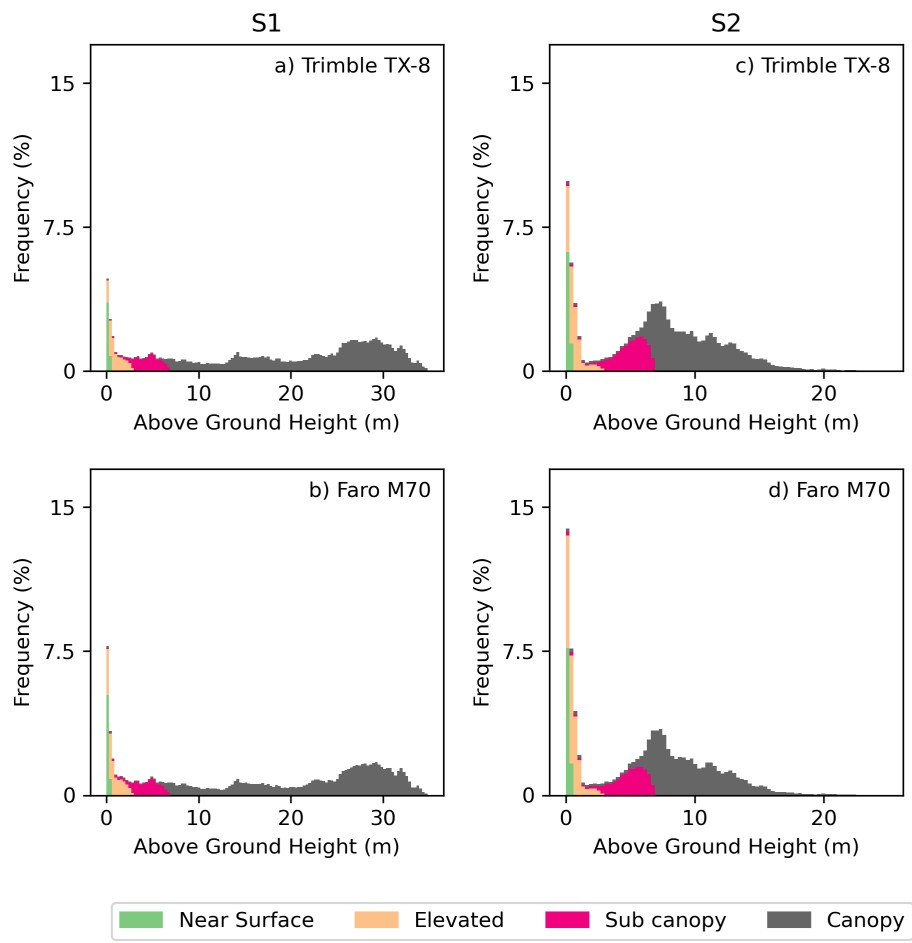

**Figure 3.** Above ground height distribution of filled voxels from the respective fuel layers within S1 (**a**,**b**) and S2 (**c**,**d**) plots captured by the Trimble TX-8 and Faro M70 scanners. Space originating from near surface is is shown in blue, elevated fuel in orange, sub-canopy in green and canopy in red.

### *3.2. Single Scan Plot Characterisation*

### 3.2.1. Noise Detection

The mean signal to noise of single scans collected with the Faro M70 scanner (91:1 at S1 and 371:1 at S2) was found to be significantly higher than the that of the Trimble TX8 scanner (10,000:1 at S1 and 60,000:1 at S2). The few noise points identified in the Trimble TX8 scanner were typically isolated and could be detected using the proposed filter (MCC = 0.6). In contrast the noise in the Faro M70 typically occurred as a spray of points following the interception of a large feature (branch or stem). These points were closely clustered with vegetation points and were more difficult to identify (MCC = 0.2).

Whilst the Faro M70 showed lower omission rates overall, the pattern of omission over distance at each of the two sites was similar. This suggests that characteristics of each site, and where occluding material is distributed, has more of an effect on omission rates than the properties of the scanners used (Figure 4). The Faro M70 scanner had higher commission rates, particularly in areas close to the scanner. The higher commission rates can be directly attributed to the increased noise present in the M70 scans.

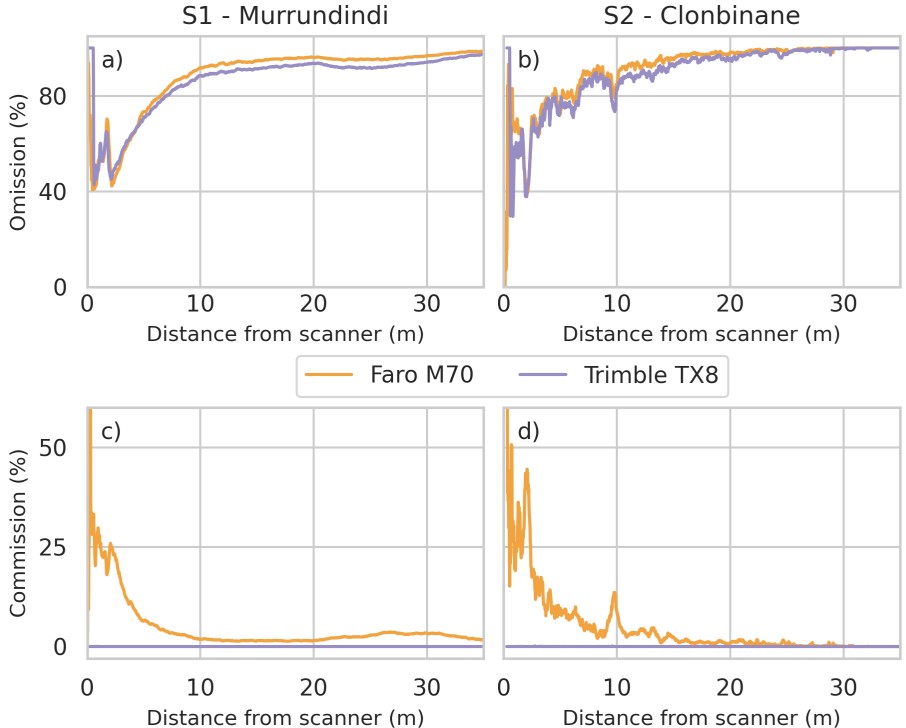

**Figure 4.** Effect of distance from scanner on the rates of omission (**a**) S1 and (**b**) S2, and commission (**c**) S1 and (**d**) S2, of the observed space for single scans when compared to the reference voxel space created using the merged scan.

### 3.2.2. Ground Definition

The terrain representation by both scanners were similar at both sites. Both scanners returned a similar number of ground points with an average of 13% and 14% of the total plot area ground being detected at S1 by single scans by the TX8 and M70 scanners respectively. At S2, an average of 4% of the ground within the total plot area was detected in individual scans. The rate of ground points present in any individual scan decreased rapidly with distance from the scanner (Figure 5). The ground filtering approach implemented in the CSF was able to detect ground points present with similar rates of omission from the two scanners. Commission points in Faro M70 scans, however, was significantly higher at the Clonbinane site. Through visual inspection the erroneously detected ground points were primarily noise points occurring below the ground and not detected by the noise

filtering algorithm. The dense covering of bracken contributed to the production of these noise points within these scans.

The RMSE when compared to reference terrain model was 0.03 m for both scanners at S1 and 0.03 m at for the TX8 and 0.04 m for the M70 scanner at S2. Small increases in difference were found with increasing distance from the scanner. These could primarily attributed to increased error where the interpolation distance increased with occlusion.

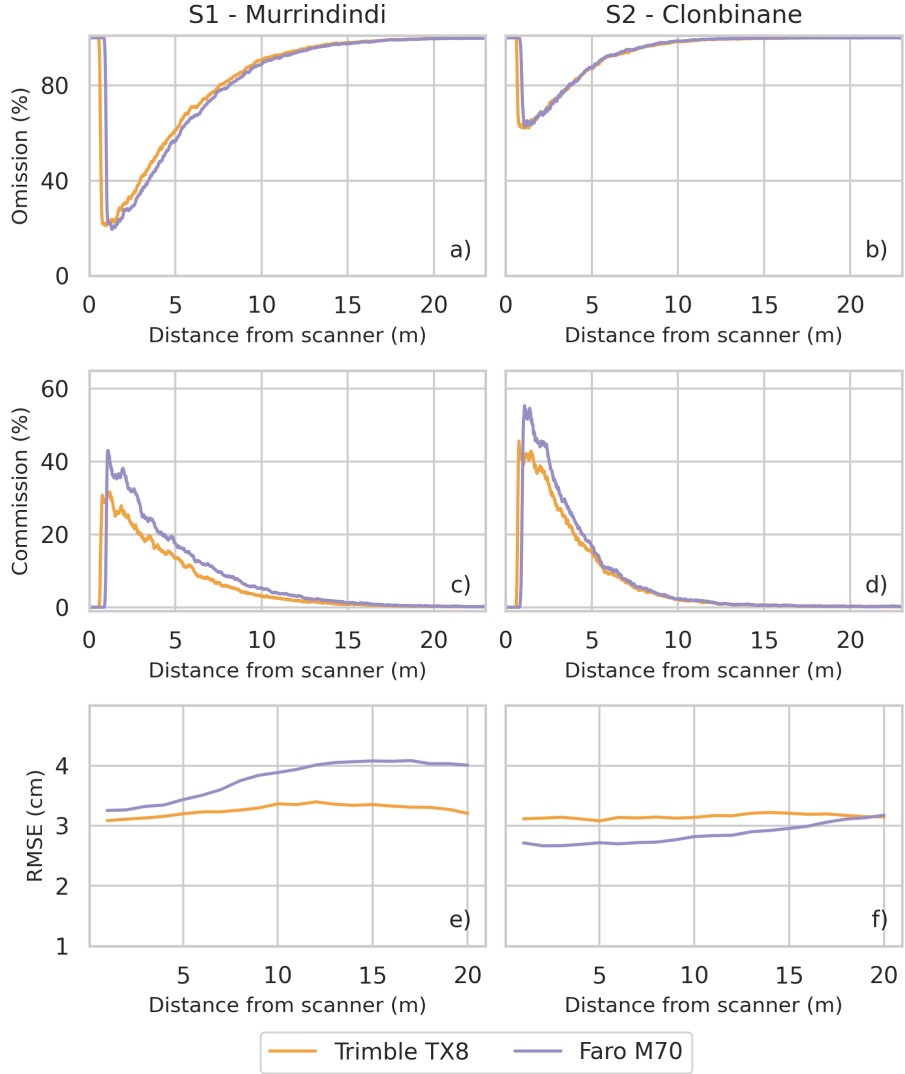

**Figure 5.** Effect of distance from scanner on the rate of ground observed when captured from a single scanner. Omission (**a**,**b**), commission (**c**,**d**) and RMSE (**e**,**f**) are shown for the single scan data in comparison to the merged scan voxel space.

### 3.2.3. Metric Evaluation

The near-surface layer metrics were well determined (within 10% and 10 cm) by most single scans at both sites and by both scanners. At both sites the absolute mean difference between the single scan and reference estimates across the 25 scans was less than 3% for cover and 0.05 m for height when considering a 4 m radius. Near-surface cover was over estimated by the TX8 scanner (by 1% at S1 and 4% at S2) and under estimated by the M70 scanner (by 2% at S1 and 3% at S2). At S1, near-surface fuel captured within 4 m of the scanner were made with the lowest RMSE in both cover and height Tables 3 and 4. At S2, similar RMSE of cover were retrieved at 4, 6 and 8 m, while near-surface fuel height was best estimated at 4 m by both scanners. Whilst elevated fuel height and cover was estimated with higher uncertainty, the general trend was similar pattern to that of near

surface fuel cover, with a sample radius of 4 m providing the best representation at both sites for both scanners.

Anomalous estimates of near-surface and elevated fuel were recorded at some scan locations at both sites (difference up to 21% in cover and 0.09 m when compared for a 4 m radius). These were affected by near scanner objects, reducing the effective observed area or breaking vertical connection of elements from other layers. This resulted in the estimates being made over an area not representative of the area at the radius around the scan (as captured by other scans in the reference scan). Near scanner objects also resulted in elements from one layer being wrongly attributed to either of the near-surface and elevated layers due to breaks in vertical connectivity.

**Table 3.** RMSE of fuel layer cover estimates from single scan data in comparison to the estimates derived from reference 3D model within the given radius of the single scan location.

| Site | Scanner<br>Radius from Scanner | Trimble TX-8 | | | | Faro M70 | | | |
|---|---|---|---|---|---|---|---|---|---|
| | | 2 m | 4 m | 6 m | 8 m | 2 m | 4 m | 6 m | 8 m |
| S1 | Near-Surface (%) | 19 | 6 | 9 | 8 | 19 | 7 | 8 | 8 |
| | Elevated (%) | 15 | 13 | 19 | 23 | 14 | 8 | 13 | 18 |
| | Sub-Canopy (%) | 19 | 7 | 6 | 5 | 11 | 7 | 5 | 7 |
| | Canopy (%) | 29 | 24 | 20 | 21 | 35 | 32 | 29 | 31 |
| S2 | Near-Surface (%) | 15 | 9 | 9 | 10 | 14 | 8 | 8 | 10 |
| | Elevated (%) | 17 | 17 | 23 | 24 | 8 | 9 | 18 | 19 |
| | Sub-Canopy (%) | 11 | 6 | 5 | 4 | 7 | 11 | 6 | 6 |
| | Canopy (%) | 18 | 12 | 10 | 9 | 11 | 13 | 10 | 9 |

**Table 4.** RMSE of fuel layer height estimates from single scan data in comparison to the estimates derived from reference 3D model for the same area.

| Site | Scanner<br>Radius from Scanner | Trimble TX-8 | | | | Faro M70 | | | |
|---|---|---|---|---|---|---|---|---|---|
| | | 2 m | 4 m | 6 m | 8 m | 2 m | 4 m | 6 m | 8 m |
| S1 | Near-Surface (m) | 0.07 | 0.05 | 0.06 | 0.07 | 0.07 | 0.03 | 0.04 | 0.04 |
| | Elevated (m) | 0.34 | 0.05 | 0.07 | 0.08 | 0.36 | 0.08 | 0.10 | 0.09 |
| | Sub-Canopy (m) | 0.44 | 0.30 | 0.35 | 0.21 | 0.35 | 0.26 | 0.24 | 0.22 |
| | Canopy (m) | 5.14 | 5.66 | 5.23 | 4.58 | 7.16 | 7.38 | 6.15 | 6.29 |
| S2 | Near-Surface (m) | 0.05 | 0.05 | 0.09 | 0.11 | 0.04 | 0.04 | 0.06 | 0.07 |
| | Elevated (m) | 0.15 | 0.08 | 0.11 | 0.11 | 0.07 | 0.06 | 0.07 | 0.09 |
| | Sub-Canopy (m) | 0.52 | 0.32 | 0.13 | 0.14 | 0.55 | 0.33 | 0.20 | 0.20 |
| | Canopy (m) | 1.33 | 1.20 | 1.45 | 1.04 | 1.56 | 1.59 | 1.49 | 1.39 |

Overall, increased sampling radii provided better representations of the upper fuel layers (sub-canopy and canopy). This is primarily due to the disproportionate effect near field occlusion had on the results from lower sampling radius. For example, of the estimates from the TX-8 scans at S1 at a 4 m radius, 22 of 25 scan locations had a cover estimate error of less than 5%. The remaining scans had cover estimates with an underestimation of cover greater than 21%. Aside from these scans the estimates of cover and height were closer to the reference scan at 4 m than any other sampling radius for canopy cover and height.

Sub-canopy cover and height were estimated without large biases in cover (a mean underestimation of cover by 2% and 5% (TX-8) and 2% and 7% (M70) at sites 1 and 2 respectively) at a sampling radius of 4 m. However, this increased to up to 17% at an 8 m sampling radius. Canopy cover was similarly underestimated at all radii. For example, at a 4 m sampling radius canopy cover is underestimated by 15% and 8% (TX-8) and 2% and 9% (M70) at sites 1 and 2 respectively. Underestimation is caused by unaccounted occlusion in the estimation of cover in this layer.

## 4. Discussion

This study illustrates an operational sampling methodology for the use of TLS to measure fuel hazard metrics within the current operating paradigm employed by land management agencies in South-Eastern Australia. In the case study environments, non-overlapping individual scans are shown to provide reliable estimates of fuel cover and height at the near surface, elevated and sub-canopy strata, without the need for co-registration. This methodology effectively balances efficiency of capture and efficacy of data. The greater precision in height and cover estimates from point-cloud based assessments can be used to support a more robust determination of overall fuel hazard at the time of assessment and monitoring over time. Whilst the case studies and associated method in this manuscript have been developed in South-Eastern Australia it is believed that as fuel metrics describing cover and height are utilised across the world in models, simulators and field based assessment methods [7–12] that the method is widely applicable but should be tested further to observe interactions between vegetation density and data efficacy.

Based on the analysis completed in this study, and other similar recent studies [30,45,46], it can be suggested that the use of TLS to observe the entire fuel complex within an existing operational framework (such as the OFHAG [6] or VESTA [7]) remains difficult. Challenges remain in the accurate separation of points originating from different fuel strata and the consistent quantification of metrics describing these layers across environments and hazard classes. Whilst we did not attempt to describe the bark-fuel strata in this study, previous research has shown that the derivation of species [47] are derivable from similar 3D point cloud based data. Therefore, fuel metrics derived from point clouds can be used to support an overall fuel hazard assessment but need to be paired with moisture assessments and surface and bark hazard scores to gain a full realisation of the overall fuel hazard.

The two scanners used in this study provided very similar descriptions of the environment, despite the differences in operating principles and cost. The primary difference between the TX-8 and M70 scanners was the noise characteristics of the produced point cloud. Noise within the TX-8 point clouds was primarily found to be spurious points located in the space between vegetation (Figure 6a). As this noise presented as points that were often isolated from vegetation elements the kNN approach to outlier detection was successful in detecting it. Whilst these spurious points were also present in the Faro M70 point clouds, a more significant source of point cloud noise was found to occur in clusters both close to and away from vegetation elements (Figure 6b). Similarly, Stovall and Atkins [32] found that the Faro phase-based scanner had substantially more noise than the Leica BLK360 in the measurement of tree characteristics. Stovall and Atkins [32] attributed the increased noise profile to the phase-based system of laser scanning which can cause ranging errors from partial interceptions, such as small branches or leaves and edges of larger trunks. The clustered nature of this noise means that the outlier detection strategy used here was unsuccessful. Several other outlier detection approaches were trialled (including one class Support Vector Machines, dbscan and isolation forests). However, as this noise appears similar to fine vegetation elements no approach provided high accuracy in the detection of these points. If a merged scan strategy is employed and thereby achieving a relatively low occlusion rate an opportunity exists to detect and remove this noise through analysing connection of point clusters to the ground. However, in a single scan approach where occlusion is high the detection of noise with characteristics presented in the Faro M70 can not be easily achieved with consistent results.

Furthermore, when a merged scan or a small number of single scans are being used, parameters for noise reduction can be fine-tuned for optimal results. However, this same approach cannot be used when trying to develop an operational method that is broadly applicable across multiple single scans, captured in varying forest sites and executed by non-specialist users. Rather, the approach developed within this paper is set within an operational paradigm and therefore approaches to processing must be consistent.

Whilst fuel metrics derived from TLS have the potential to provide a greater level of precision, there are a number of limitations which need to be considered when using

TLS. Firstly, surface fuel, fuel laying horizontally on the ground, appears very similar to the ground itself within 3D space. The ground filter used in this study was originally developed for airborne laser scanning data sets and includes a parameter that sets all points within a distance to the ground as originating from the ground. This value was set to 0.03 m. Thus it is likely a significant proportion of classified ground points originate from this layer. It is for this reason that no attempt was made to quantify surface fuel height and cover in this study.

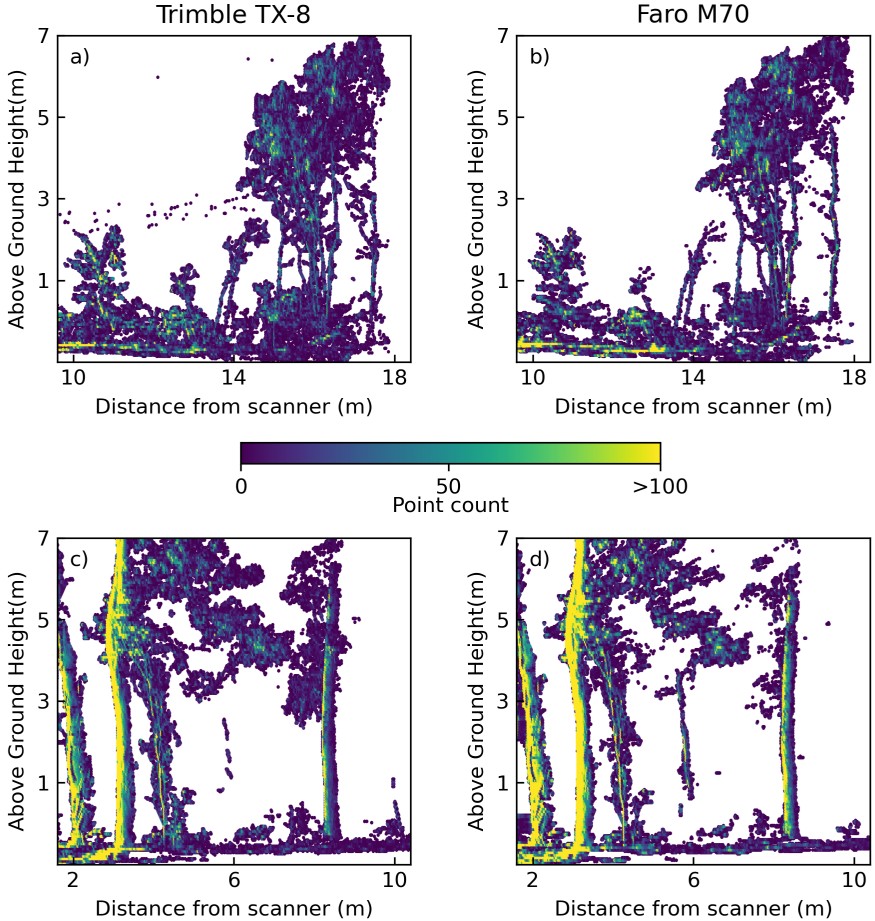

**Figure 6.** Examples of the characteristic noise in (**a,c**) Trimble TX8 and (**b,d**) Faro M70 scans. Panel (**a**) shows several isolated noise points between 2 and 6 m from the scanner and at a height of 2 to 3 m. Panel (**d**) shows the characteristic noise of the Faro scanner where noise extends in the direction of the emitted laser prior to the object.

A second limitation exists in the observation of canopy cover. Canopy cover estimates from single scan TLS point clouds in this study were highly variable. This is consistent with findings presented by García et al. [48], Srinivasan et al. [49] who also highlighted variability in canopy cover estimates due to overshadowing and angular resolution. However, when estimating fuel hazard in the context of the current operating paradigm in South-East Australia, the measurement of canopy cover is not essential and therefore we do not consider it to be a major impediment to the use of TLS by fire management agencies. However, in forest types where canopy cover or canopy fuel load estimates are required (e.g., Mallee fire behaviour models [50], the Pine Plantation Pyrometrics [51]) then it is suggested that the deployment of above-canopy sensors will be necessary. For example low-cost sensors onboard drones have been shown to be able to determine canopy properties reliably [29,52,53]. Previous literature however has demonstrated that canopy base height measurements, which are used in crown fire models across the world [10,54,55], are able

to be derived accurately from TLS point clouds [48]. Furthermore, TLS point clouds in conjunction with novel leaf and wood separation algorithms have illustrated the potential for fine fuel to be separated from woody material in eucalypt forests [30].

Therein lies the challenge of rationalising the potential of advanced metric derivation from point clouds with existing methods of fuel hazard assessment. A key reason for the vertical stratification of fuel into layers is to provide a simple and consistent framework within which observations can be made. The description of these layers then allow for inference describing the connections and distribution of fuel within the entire complex ultimately for the use in risk and behaviour modelling. However, having defined layers can lead to confusion during visual assessment especially in vegetation that does not conform with distinct layer definitions or with assessors who may not be experienced in assessing fuel hazard [14,15,56]. Furthermore, point cloud analysis with defined height thresholds is difficult in environments where vegetation may span across two or more height thresholds (as seen at S2 in Figure 7). On the other hand, 3D point clouds such as those provided by TLS provide a data source that inherently represent the connection of the fuel within the surrounding area without the need to derive aerial metrics such as mean height and cover. The challenge this presents both the fire and remote sensing research community is how to leverage this rich data source to provide an actionable set of information for use in operational fire management activities. Therefore it is suggested that future approaches look to utilise dynamic layering and to generate structural metrics that describe vertical connectivity including the presence and density of ladder fuels whilst also being linked to fundamental fire behaviour properties [57,58]. Further research opportunities also exist in the determination of bark hazard from remote sensing. By leveraging 3D information within the point clouds, fire managers are able to understand accurately determine fuel conditions at the time of assessment and also identify what and why things might be changing in future assessments with the ability to compare and reanalyse 3D information into the future.

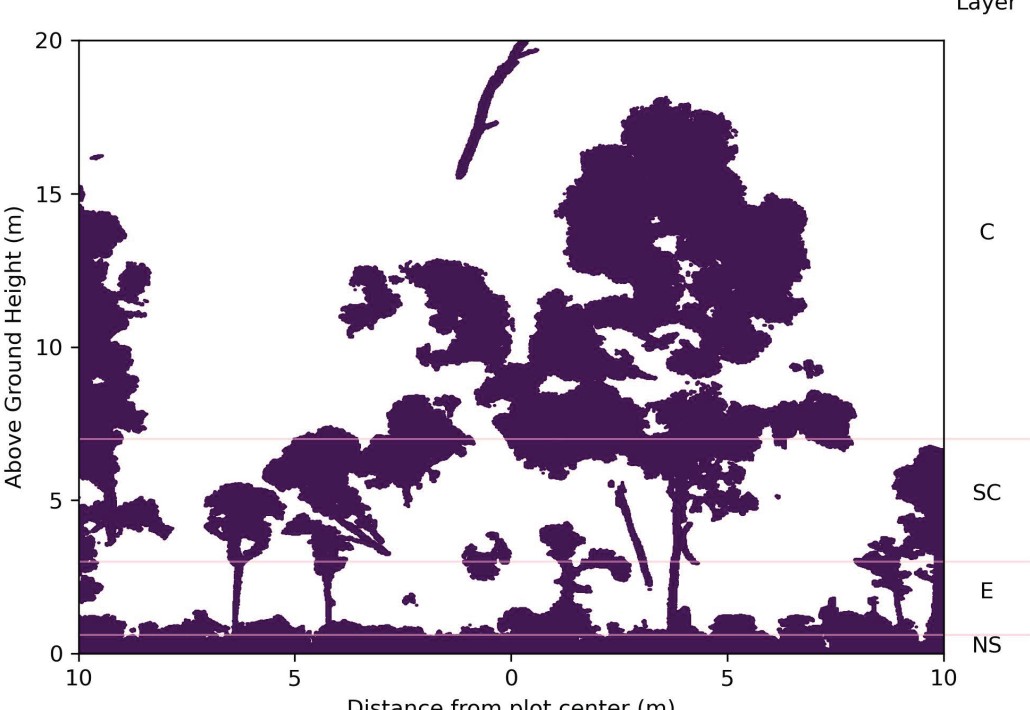

**Figure 7.** A hexbin plot of a 1 m transect of the reference point cloud taken through the middle of the plot established as S2 (Clonbinane). The defined height of fuel strata are shown as pink lines.

Whilst it is shown that TLS point clouds can describe vegetation structure, it is pertinent to note that the two scanners tested in this manuscript do not provide information on

non-morphological traits pertinent to fire behaviour. Multi-spectral LiDAR sensors have demonstrated potential to extract vegetation physiological traits such as NDVI profiles [59], chlorophyll concentration [60] and vegetation water content [61,62]. The ability to capture both structural and fuel moisture estimates from multi-spectral LiDAR has the potential to provide a complete picture of the above-ground fuel risk across the profile and augment current remotely sensed estimates of fuel moisture derived from satellite observations. Given the cost of multi-spectral sensors, difficulty in aligning sensor data and the assignation of moisture values to fine scale vegetation objects, it is unlikely that these sensors will rapidly become operationally viable for land and fire management agencies. Despite this, the structural information contained in single-wavelength scanner data can assist in fuel moisture modelling which has found strong relationships between vegetation structure and dead fine fuel moisture content beneath the forest canopy [63–65].

## 5. Conclusions

The method presented in this manuscript represents a step forward in the viability of TLS to be used in an operational paradigm to support overall fuel hazard assessments. We conclude that individual non-overlapping TLS scans capturing and clipped to a 4 m radius area offer similar estimates of fuel coverage and height at the near-surface and elevated and sub-canopy strata to the full plot. These assessments can then be used to support a more robust determination of overall fuel hazard at the time of assessment and monitoring over time. This approach balances time efficiency in both the capture and processing phases of analysis, as well as data efficacy. Furthermore, it was shown that the lower-cost instrument (Faro M70) used in this study provides comparable terrain and vertical and horizontal vegetation information to that of a higher-cost instrument (Trimble TX-8). We also found that there are limitations to the use of TLS to measure fuel hazard, namely in the determination of surface fuel hazard, bark hazard and the analysis of live and dead fuel. We suggest that for the full capability of TLS to be realised operationally, methods that leverage the full description of the vegetation provided by TLS be used. Further research should also focus on field-based and other remote sensing approaches to be developed in concert with structural measurements to facilitate the collection of fuel attributes not able to be measured by using TLS, e.g., moisture content. Nevertheless, the operational method presented in this manuscript for the measurement of fuel from TLS provides an incremental step forward for fire managers looking to utilise a non-subjective, more precise estimate of fuel.

**Author Contributions:** Conceptualisation, L.W., S.H. and R.T.; data curation, L.W., B.H. and A.W.; formal analysis, L.W. and S.H.; funding acquisition, L.W. and S.H.; methodology, L.W., S.H., R.T. and J.M.; resources, S.H.; software, L.W., S.H. and B.H.; writing—original draft, L.W. and S.H.; writing—review and editing, B.H., R.T., A.W. and J.M. All authors have read and agreed to the published version of the manuscript.

**Funding:** This research was funded by the Department of Environment, Land, Water and Planning through the Bushfire and Natural Hazards Cooperative Research Centre with Grant Number ERP29.

**Data Availability Statement:** Data and python scripts can be made available upon request.

**Acknowledgments:** The funding support of the Victorian Government through the Bushfire and Natural Hazards Cooperative Research Centre and coordination through Simon Jones at RMIT University is acknowledged. Julian Madigan is acknowledged for his contribution of fuel subject matter expertise. The Department of Environment Land Water and Planning staff and, in particular Belinda Quinton, Mick Morely, Tom Goldstraw and Jamie Liddle are gratefully acknowledged for prioritising burn areas and providing local knowledge in accessing sites. S. Hillman acknowledges the support of the Fulbright Australia program for which SH was a recipient of whilst this work was conducted.

**Conflicts of Interest:** The authors declare no conflict of interest.

## Abbreviations

The following abbreviations are used in this manuscript:

| | |
|---|---|
| AGH | Above Ground Height |
| CSF | Cloth Simulation Filter |
| DTM | Digital Terrain Model |
| EVC | Ecological Vegetation Class |
| MCC | Mathews Correlation Coefficient |
| NN | Nearest Neighbour |
| OFHAG | Overall Fine Fuel Hazard Assessment Guide |
| RMSE | Root-Mean-Squared Error |
| ST | Selection Threshold |
| TIN | Triangular Irregular Network |
| TLS | Terrestrial Laser Scanning |
| 3D | Three-Dimensional |

## Appendix A. Overall Fuel Hazard Guide Assessments

**Table A1.** Overall fuel hazard ratings for S1 and S2, following the methodology developed by Hines et al. [6].

| Site | S1 | S2 |
|---|---|---|
| Canopy height (m) | 25 | 10 |
| Canopy cover (%) | 70–80 | 60–70 |
| Stringybark fuel hazard | High | Moderate |
| Ribbon bark fuel hazard | Moderate | Moderate |
| Other bark fuel hazard | N/A | N/A |
| Overall bark fuel hazard | High | Moderate |
| Elevated cover (%) | 20–30 | 30–50 |
| Elevated dead (%) | 20 | 20 |
| Elevated fuel mean height (m) | 0.60–0.70 | 0.5–0.60 |
| Elevated fuel hazard | Moderate | High |
| Near-surface cover (%) | 20–40 | 20–40 |
| Near-surface dead (%) | 20 | 20 |
| Near-surface mean height (m) | 0.20–0.40 | 0.20–0.40 |
| Near-surface fuel hazard | High | High |
| Surface litter cover (%) | 90 | 80–90 |
| Surface litter depth (mm) | 25–45 | 20–30 |
| Surface fuel hazard | Very High | High |
| Combined hazard | Very High | Very High |
| Overall fuel hazard | High | Very High |

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
