# Peer review of "Terrestrial Laser Scanning: An Operational Tool for Fuel Hazard Mapping?"

_fire, doi:10.3390/fire5040085_

Round 1
Reviewer 1 Report
Precise measurements of fuel hazard are important for understanding fuel dynamics and validating fuel models used in fire behavior modeling. However, the novelty of the proposed methods is limited, or the authors failed to highlight the research contributions in both the Abstract and Introduction parts. In addition, there are still several major concerns that should be clearly addressed:
1. There was too much description for fuel hazard assessment using the visual interpretation method in the Introduction part, but that is not the focus of the study. I would suggest the authors shorten or refine it.
2. The authors did not elaborate on the contributions of this paper clearly in the introduction part. Too many words were used to demonstrate the advance in the TLS’s application for forest parameters extraction, but the author didn’t provide the major difference and improvement of the proposed method.
3. The aerial and mobile laser scanning can also provide a high-density 3-D point cloud with high precision, and the mobile laser scanning is more convenient to move in the environment, so why not map the fuel hazard using them?
4. There was no validation data information found. Thus, how to validate the accuracy of the height and cover and ensure the accuracy of the field data.
5. In the TLS data capture part, I would suggest the author provide the accuracy of the co-registration.
6. The specific locations of the plots used should be highlighted in Figure 1a.
7. The authors stated a method for the fuel hazard assessment, but the novelty of the methods is limited, and many existing methods were used in the Methods part. The authors should shorten the description of the existing methods and highlight the novelty of the method.
8. Why not estimate the height and cover using the method proposed for the forest structural parameters mentioned in the literature, but develop a method to estimate the height in hazard metrics. What is the major difference or the improvement in your study compared with other height or cover estimation methods?
9. The legend in Figure 2 is suggested to be placed in an obvious position, for example, the middle position. The labels of the horizontal axes (distance from scanner) in Figures 4 and 5 are suggested to be placed in corresponding figures, respectively.
10. What do the different colors represent in Figure 6 and Figure7? There was no legend in Figure3, Figure6, and Figure7.
11. What is the ball shaped structuring element in line 239? I suggest the full name of the RMSE should be placed in the position that first appears.
12. The Result part is confused, and the titles of the sections should correspond to the titles of the method sections as far as possible.
13. In the Conclusion part, it would be better to point out the main contributions of the proposed method and then conclude the new findings from the results and analysis.The sentences in the Conclusion part should be refined.
Reviewer 2 Report
This study is poorly designed and not meeting the operations goals as set in the objections. Results from TLS scanning in two plots cannot be generalized over a landscape. The manuscript is poorly written with unnecessary texts and length. Methods and results that are not relevant to the study goals are presented.
Please see my comments embedded in the manuscript.

Reviewer 3 Report
Summary:
The authors of “Terrestrial Laser Scanning: An operational tool for fuel hazard mapping” propose a basic method or framework for measurement of fuel heights using TLS in order to remove subjectivity from sampling efforts. The authors point out that gains from accuracy must be balanced with sampling effort in data collection and time consuming processing of point clouds by including input from land managers/fire practitioners. Additionally, they point out that it must be cost effective. In doing so, they land on a single scan methodology that is becoming the consensus in the TLS monitoring community. The objectives were to: provide guidance to an appropriate sampling strategy, compare low and high cost TLS solutions, and to evaluate fuel strata metrics. The authors conclude that single scans from a low cost laser can produce comparable results to registered point clouds from high-end research grade lasers, but with limitations from noise and range issues. While nothing novel is presented in the manuscript (a new method for measuring fuels, a unique TLS workflow or data process, etc.) the authors show that the two units being compared have limited utility in generating precise objective fuel depths that can, presumably, be used in conjunction with existing subjective bark estimations to improve efficiency in assessing fine fuel hazard risk in New Zealand.
Major issues:
There are a few major and minor issues of this study and all were related to the three objectives: the framework provided is limited to a few unique hazard metrics used in New Zealand that are based on structure metrics not used extensively internationally (OFHAG), only two systems were compared and limitations of each system conflated cost with laser type, and measurements were not evaluated with actual OFHAG metrics.
The authors develop a framework that, while partially useful to fuel hazard metrics in New Zealand and is too specific for more extensive use. This is a minor point, but it
While some comparisons of TLS systems are in the literature, i.e. Stovall and Atkins 2021 (doi:10.20944/preprints202107.0690.v1), this study compares an inexpensive phase-difference vs. expensive time of flight. It is noted that the limitations documented from the respective noise issues from each type are indicative of their type and power, and not necessarily cost as there are representatives from each type across the cost spectrum and an expensive $100k USD+ phase laser will have the same noise issues as the low cost ones without extensive “black box” proprietary software noise algorithms that are beyond the technical reach of most land managers.
Additionally, this reviewer is concerned that the evaluation of the TLS systems were compared against a reference of the registered TX-8 scans, presumably to represent the true fuels structure and not against the method described in section 2.2. In fact, although overall hazard was assessed (lines 153-156), no mention of the actual assessment is mentioned in the results or discussion that is discernable from the reference registered point cloud from the TX-8. There are many issues with taking an object based approach in fine fuels from multiple registered point clouds that mere voxelization can rectify, especially at a 2 cm (8 cm2?) resolution. Fine fuel occlusion from dense surface fuels being one and the movement of vegetation by wind with overlapping scans effectively doubling the fuel load, or cover in this case. A much more interesting manuscript would have been the adaptation of point cloud measures or metrics to the existing OFHAG methods. An example would be in the CBDI post fire evaluation by Gallagher et al. 2021 (doi:10.3390/rs13204168).
Major revisions are required to address these concerns.
Minor issues:
Lines 18-19: This first sentence is unnecessary and does not improve the manuscript. I would drop it completely.
Line 47: TLS is still young and it is early to say “gold standard”. Change this to “…increasingly being considered as a tool...”
Line 64: add “to” between “led” and “the”.
Line 119: space between includingAcacia.
Line 120-122: Edit this line for clarity; with/were also with…
Line 180: The only software listed in the manuscript was Trimble RealWorks 11.1. Presumably, the voxelization, noise detection, normalization and data analyses were done in other software/packages. Please cite them.
Line 181-182: Insert "that" in between "confirm" and "the accuracy..."
Line 196-198: The method you are using here to use a registered point cloud as a reference to evaluate both TLS systems must first be evaluated for fine fuels or a reference should be cited that has validated this method.
Line 200: change “by” to “be”
Line 221: comma after “parameters”
Line 283: Rates of omission and commission are comparisons to the invalidated reference point cloud correct? Points that avoided noise reduction from the method described, may escape reduction and generated false omission and points that were falsely classified as noise could give false commission.
Line 308: S2 is shown in figure 2b and 2d.
Line 309: The notable increase in points is just below 5 meters, not 7 and trails off, if 2d is what the authors are referring to.
Line 312: S1 is shown in figure 2a and 2c.
Line 314: this is a discussion point. State only results in this section.
Line 323: Heights and percent covers alone are not fuel hazard characterizations. These are simply mean and standard deviations of points in a point cloud and percent cover of voxels in a point cloud. Fuel hazard metrics that can be interpreted by land managers are conflated throughout the manuscript with raw point cloud metrics that require interpretation. These percent cover values from voxels have not been validated with actual cover estimates. In some cases, single grass culms could make up voxel spaces that overestimate cover.
Lines 469-492: This is excellent
Lines 520-522: The specific low cost unit that you tested. The system that you tested has been shown to create significantly more noise than other low cost units. Stovall and Atkins 2021 (doi:10.20944/preprints202107.0690.v1)
Round 2
Reviewer 1 Report
The revision meets my expection, thanks very much for authors' effort and time.
Author Response
The authors thank the reviewer for their time in reading and reviewing the manuscript and appreciate their feedback in helping to improve the manuscript.
Reviewer 2 Report
Reduce lengthy texts in the background on TLS.
Write a justification on why only two plots are sufficient to meet your objectives.
Add a reference on how these voxel grids are produced? and what those grids represent?
Reviewer 3 Report
Review report Round 2
The authors have resolved most of the major issues sufficiently for publication with a few minor edits detailed below.
Authors:The authors would like to thank the reviewer for their time taken to review the manuscript. Their comments and suggestions of improvement to the manuscript are valuable. The authors have responded to each comment and amended the manuscript where appropriate. As an initial point of clarification, the authors would like to point out that the case studies were conducted in South Eastern Australia.
Reviewer response: Apologies, I was reviewing a study in New Zealand at the same time and must’ve gotten my signals crossed. I hope that I didn’t write South Eastern Australia in that one… Fortunately, the rest of the comments relate only to your study.
Authors: The authors thank the reviewer for their comment and would seek to highlight that despite the case studies being conducted in South Eastern Australia, the fuel metrics of cover and height of below-canopy strata are common across fire prone countries across the world. To address this comment, the authors have added the following text in the introduction text in the introduction. Line 37 Similar fuel attributes are measured across the world that help to inform fuel dynamics understanding. For example, in the USA, the Fuel Characteristic Classification System is a comprehensive guide where fuel beds are described by firstly identifying the strata and subsequently assessing a number of variables (e.g. cover, height, density) [Prichard et al., 2013]. Cover and height metrics are used both directly and indirectly through the determination of fuel load in fire behaviour models and simulators around the world(US: [Rothermel, 1972], Canada: [Group, 1992], Australia: [Gould et al., 2008, Tolhurst et al., 2008], Australia and Europe: [Anderson et al., 2015])
Reviewer response: Yikes, again with New Zealand. My bad. Anyway… Maybe, but they are measured differently across all of those attributes requiring modeling against physically collected data at each location before raw point cloud metrics may be applied. It is clearly outside the scope of this study with such a small sample size, but it seems like not a lot can be determined with the current data.
Authors: The authors acknowledge the reviewers concerns and highlight the statement above in relationship to the applicability of the metrics derived from point clouds in their applicability across the world. The authors have added in the following statement in the discussion: 2 Line 400 Whilst the case studies and associated method in this manuscript have been developed in South Eastern Australia it is believed that as fuel metrics describing cover and height are utilised across the world in models, simulators and field based assessment methods [Prichard et al., 2013, Rothermel, 1972, Group, 1992, Gould et al., 2008, Tolhurst et al., 2008, Anderson et al., 2015] that the method is widely applicable but should be tested further to observe interactions between vegetation density and data efficacy.
Reviewer response: Wow, I promise that I don’t normally lump Australia/New Zealand/Tasmania into one country and recognize the differences. Please excuse my continued errors. The key line of “tested further” appeases me.
Authors: The authors thank the reviewer for their comments and agree with them that cost alone is not solely a separation between the two scanners, rather are indicative of their type and power. To this extent, the authors have refined the objectives of the paper. Line 90 From: 1. To provide guidance as to an appropriate sampling strategy for use in the collection of TLS data; 2. To compare two low and high cost Terrestrial Laser Scanners for fuel hazard assessments; and . To contrast the uncertainty within observations made within each fuel strata. To: 1. To provide guidance as to an appropriate sampling strategy for use in the collection of TLS data; 2. To compare two Terrestrial Laser Scanners with different operating principles and price points for fuel hazard assessments.
Reviewer response: Fair enough
Authors:There are very few options for providing a complete and accurate representation of fuel hazard in a plot such as this. We consider merged TLS data that has been manually edited as the best and most complete representation of fuel possible within the constraints of the project. The reference scan was collected in the best possible conditions and thoroughly checked for noise and misregistration by a skilled operator familiar with point clouds and the conditions at the site. The 2x2x2 cm voxels could be seen to capture most if not all of the movement and residual misregistration between scans. Nevertheless we agree that this is not a perfect representation of the site, just the best that could be achieved. We have now included Overall Fine fuel hazard Assessment Guide (OFHAG) values within Table 2 and the full OFHAG output in Appendix 1. Whilst adapting TLS to OFHAG outputs (i.e. achieve an overall hazard assessment) may be of interest we feel that this would be misleading as TLS data cannot represent some of the attributes that are key to this assessment. We cover this in the discussion. Approaches such as RF used in Gallagher et al. [2021] may provide reasonable outcomes but would be strongly linked to the amount of data. Our paper is perhaps a forebearer to such a work as it provides guidance to implement TLS in an operational manner and as such an avenue increase the available data to do this analysis To this end, we have added the following text to highlight the key findings of our study and the relationship to fuel hazard assessments. . 5 Line 392 This study illustrates an operational sampling methodology for the use of TLS to measure fuel hazard metrics within the current operating paradigm employed by land management agencies in SouthEastern Australia. In the case study environments, non-overlapping individual scans are shown to provide reliable estimates of fuel cover and height at the near surface, elevated and sub-canopy strata, without the need for co-registration. This methodology effectively balances efficiency of capture and efficacy of data. The greater precision in height and cover estimates from point-cloud based assessments can be used to support a more robust determination of overall fuel hazard at the time of assessment and monitoring over time. Whilst the case studies and associated method in this manuscript have been developed in South Eastern Australia it is believed that as fuel metrics describing cover and height are utilised across the world in models, simulators and field based assessment methods [Prichard et al., 2013, Rothermel, 1972, Group, 1992, Gould et al., 2008, Tolhurst et al., 2008, Anderson et al., 2015] that the method is widely applicable but should be tested further to observe interactions between vegetation density and data efficacy. Line 405 Based on the analysis completed in this study, and other similar recent studies [Rowell et al., 2016, Gupta et al., 2015, Hillman et al., 2021], it can be suggested that the use of TLS to observe the entire fuel complex within an existing operational framework (such as the OFHAG [Hines et al., 2010] or VESTA [Gould et al., 2008]) remains difficult. Challenges remain in the accurate separation of points originating from different fuel strata and the consistent quantification of metrics describing these layers across environments and hazard classes. Whilst we did not attempt to describe the bark-fuel strata in this study, previous research has shown that the derivation of species Othmani et al. [2015] are derivable from similar 3D point cloud based data. Therefore, fuel metrics derived from point clouds can be used to support an overall fuel hazard assessment but need to be paired with moisture assessments and surface and bark hazard scores to gain a full realisation of the overall fuel hazard.
Reviewer response: This is still a sticking point with me. The same reason that many researchers have moved away from MLS in fuel sampling (the difficulty of separating noise from vegetation points due to the inherent decimation in SLAM for MLS) is analogous to the reason you can’t register scans from TLS (noise from vegetation movement at voxelized scale and an artificial increase in vegetation points). Many of the researchers you sited in support of registration of multiple scans have moved away because registration creates less accurate point clouds and you are using them as reference. As someone who has registered hundreds of point clouds, this is suspect. How do 25 scans across 400m2 not look like an absolute blurry mess especially after voxelization? However, it is the best you can do with what you have.
Authors: The authors have added in a statement to highlight that a set of in-house python scripts were used to process the data. We have added a statement in the data availability section to make this available upon request. Line 156 Point cloud processing and analysis was completed using Python code that was developed in-house specifically for TLS point cloud processing. The following steps were undertaken.
Reviewer response: The data availability section in the latest revision says that the data is available, but does not mention the Python scripts/code. Please add this.
Authors: The reviewer is correct that there are potential risks to using this approach. Nevertheless, we contend that the manually cleaned multi-scan point cloud provides the most complete noise free representation of the environment possible beyond simulation. This approach has been previously used in Hillman et al. [2021]. As part of the refinements to the manuscript, the authors have made the following change:
Reviewer response: Fair enough
Author: The reviewer is correct that the omission and commission are determined against the reference cloud. There are number of limitations to this approach including those pointed out by the reviewer. Nevertheless, we contend that the manually cleaned multi-scan point cloud provides the most complete and noise free representation of the environment possible beyond simulation and that the rate of these false classifications is likely to be minimal and balanced. Nevertheless we have purposefully designated these as rates rather than error and double checked the manuscript to ensure this is the case.
Reviewer response: I contend the opposite, however, let’s agree to disagree as there is nothing else that you can do and I feel that this study has merit and should not be held back due to my opinion.
Authors: Line 312: S1 is shown in figure 2a and 2c.
As per the comments made by reviewer one - figure 2 has been updated.
Reviewer reponse: This is still not correct. Line 386 states that S1 is 2d. It should be 2c, correct?
Authors: Line 323: Heights and percent covers alone are not fuel hazard characterizations. These are simply mean and standard deviations of points in a point cloud and percent cover of voxels in a point cloud. Fuel hazard metrics that can be interpreted by land managers are conflated throughout the manuscript with raw point cloud metrics that require interpretation. These percent cover values from voxels have not been validated with actual cover estimates. In some cases, single grass culms could make up voxel spaces that overestimate cover.
The authors thank the reviewer for bringing up a pertinent point into the utility of the fuel metrics derived from TLS point clouds. The authors seek to clarify two main points. Firstly, the accuracy of the volume and cover estimates derived from 3D point clouds has previously been validated elsewhere in the literature. To address this point, the authors have added in the statement within the methods: 11 Line 267 Furthermore Hillman et al. [2019], Rowell et al. [2020], Hudak et al. [2020], Loudermilk et al. [2009], Cooper et al. [2017] demonstrated a high level of accuracy between fine-scale volume and structural measurements derived from TLS and laboratory and field measurements of biomass and fine fuel properties. Therefore, in the context of this case study, the merged TLS scan derived from the Trimble TX-8 is considered to provide the source of validation. Whilst Wallace et al. [2016] showed that repeat measurements of and environment can provide high precision estimates of change within an environment following fire. Furthermore, based on comments from reviewer 1, we have added in the visual fuel hazard scores within the results section in table 2 and appendix 1.
Reviewer response: It is agreed that these have been covered in the literature, however in most/all cases a significant amount of modeling is needed to make those connections. In the case of Hudak et al., there is a warning in the discussion that “efficient destructive sampling is required to construct a robust empirical model needed to convert occupied voxel counts to physical estimates of fuel bulk density”, and indeed, Rowel et al. and Loudermilk et al. do the same for cover and biomass. As far as I can tell, no modeling has been done to link your TLS to these metrics in your fuel type, and with so few sites, no modeling can be done. However, adding the fuel hazard scores allows the reader to compare the methods albeit in a limited way and this appeases me as well.
New comments:
Line 245. 8 cm3 resolution
Line 385 snags?
Line 386 S1 is 2c
References: Please reorder references in the order of the revised manuscript and add all missing DOI links.
